# Hepatitis B and C in Pregnancy and Children: A Canadian Perspective

**DOI:** 10.3390/v15010091

**Published:** 2022-12-29

**Authors:** Andrew B. Mendlowitz, Jordan J. Feld, Mia J. Biondi

**Affiliations:** 1Viral Hepatitis Care Network, Toronto Centre for Liver Disease, University Health Network, Toronto, ON M5G 2C4, Canada; 2School of Nursing, York University, Toronto, ON M3J 1P3, Canada

**Keywords:** hepatitis B, hepatitis C, pregnancy, pediatrics, treatment, prevention, elimination

## Abstract

In 2016, the World Health Organization released a plan to eliminate viral hepatitis as a public health threat by 2030. For Canada to achieve the recommended decreases in HBV- and HCV-related new diagnoses and deaths, an increase in services is urgently required. Identifying those at risk of, or who have acquired HBV and HCV, remains a challenge, especially with the emergence of new priority populations such as pregnant persons and children. Importantly, prenatal, and pediatric care are times when individuals are often already engaged with the healthcare system, leading to the potential for opportunistic or co-localized care and interventions. At present, Canada may not be maximizing all available virologic tools that could lead to increases in prevention, identification, improved management, or even cure. Here, we describe the continuum of care that includes preconception, prenatal, postpartum, and pediatric stages; and identify current global and Canadian recommendations, findings, and opportunities for improvement.

## 1. Introduction

In the absence of treatment, chronic infection with viral hepatitis can lead to cirrhosis and liver cancer. In 2020, 279 million people were estimated to be infected with the hepatitis B virus (HBV) infection and 59 million people with hepatitis C virus (HCV) infection globally [1]. In Canada, there are an estimated 204,000 people living with chronic HCV [2] and between 250,000 and 460,000 people living with chronic HBV infection [3]. In 2016, the World Health Organization (WHO) released a global health sector strategy to eliminate viral hepatitis as a major public health threat by 2030, providing global and national targets for reductions in HBV and HCV-related deaths, diagnoses and service coverage [4]. For both HBV and HCV, the strategy calls for 90% of those living with infection diagnosed and 80% treated by 2030 [5]. Canada has committed to reaching these targets and in response, prioritized the representation of populations disproportionately impacted by HBV and HCV, calling on provinces to develop action plans to meet the targets for viral hepatitis elimination.

Women of childbearing potential, pregnant persons, and children are emerging priority populations. In the context of HCV, new infections among women of childbearing potential have been shown to exceed men in Canada, and Canada is a major destination country for individuals from HBV-endemic areas. Pregnant persons and children are considered vulnerable populations, yet public health data have demonstrated that women do not always receive optimal prenatal care, and vaccination rates among children are decreasing. In the context of a primary care provider shortage, navigating the healthcare system may not be straightforward for pregnant persons. In fact, only 68% of Canadian women receive an appropriately timed first-trimester ultrasound, and barriers are more significant for those who are new to Canada, younger, incarcerated, or other factors [6,7]. These vulnerabilities may overlap with risks for having acquired or acquiring HBV or HCV. To prevent vertical transmission, the WHO strategy has called for routine screening of pregnant women for HBV and the scale up of universal access to HBV birth-dose vaccines to 90% by 2030 [5]. For HCV, the strategy has called for HCV testing for pregnant women at-risk [5].

Although there may be some overlap in risks for HBV and HCV acquisition among women and children, the strategies to prevent transmission, diagnose, link to care, and treat; may be distinct. This is also true of prevention of transmission to partners and children. Although this is largely a health systems issue, technological advances in prevention, diagnostics, and treatment could have a major impact on decreasing morbidity and mortality in this population. Here, we describe virologic interventions towards the elimination of HBV and HCV in Canada.

## 2. Preconception Planning

Preconception planning for HBV involves not just the individual trying to get pregnant, but the partner, as well as other caregivers [8,9,10]. Canada is a major destination country for individuals and families from regions of the world with a high rate of HBV endemicity, many of whom are above the age to have received universal birth-dose vaccination in their country of birth. At present, HBV screening is not a part of the mandatory immigration medical exam for Canada [11,12], and therefore due to its asymptomatic nature, people may not be screened until many years later, if ever, potentially leading to transmission events to close contacts as well as asymptomatic progression of liver disease. There is a lack of Canadian literature on the success of HBV preconception care. If the person becoming pregnant is negative, and there is a potential that the partner could be positive, screening the partner is indicated; but unfortunately, partner status is often unknown. If a partner is positive, immunization status should be confirmed for the uninfected partner, as sexual transmission may occur during conception [13]. Once the partner is immune, no reproductive assistance processes are required, such as sperm washing [14]; and pursuing intrauterine insemination (IUI) or in vitro fertilization (IVF) has not been shown to decrease transmission [15]. Although the threshold for HBV treatment initiation has been well described [13], there may be additional considerations in preconception planning. For example, in addition to the routine recommendations for treatment initiation, continuous HBV treatment starting preconception, even with an HBV DNA < 10^2^ copies/mL has been examined with favourable outcomes [16]. In countries where HBV is endemic, initiating antiviral treatment in the non-pregnant partner (in this study called father-to-child transmission) prior to pregnancy, may decrease transmission by ~4-fold (12.0% to 3.5% [17]), especially if the viral load in blood is greater than 10^5^ copies/mL, and in 10^3^ copies/mL in semen [17].

Due to underscreening and the lack of routine screening during immigration, close contacts such as other family members should also be screened if not known to be negative and immune in the past [13]. This is to ensure that all close contacts are immunized and to ensure the child receives birth dose vaccination should a caregiver other than the parents be positive [18,19].

Unlike HBV, HCV is curable. Clearance of HCV eliminates the risk of sexual and perinatal transmission and reduces the risk of pregnancy complications [20]. Preconception may be a period where women are motivated to make lifestyle modifications such as engaging in harm-reduction behaviours to ensure improved maternal and newborn outcomes [21,22]. In the era of high HCV cure rates, it’s important to consider the impacts of HCV in pregnancy, such as the broad relationship between HCV and ovarian senescence, risk of miscarriage, infertility, and premature birth; necessitating discussion and counselling at the preconception and family planning stage [23]. Preconception counselling and care presents the opportunity for clinicians to explain risk of vertical and horizontal transmission and the linkage of HCV infection to adverse pregnancy and perinatal outcomes and infertility. In addition, attaining HCV RNA status prior to conception with a measure of viremia can provide valuable knowledge regarding the odds of adverse pregnancy and perinatal outcomes and risk of transmission [20].

As Canadian guidelines currently recommend HCV treatment postpartum, care and counselling at the family planning stage can provide opportunity for the integration of HCV testing and decisions regarding of timing of contraception while undergoing treatment prior to pregnancy [24]. In Canada, if there is an opportunity, preconception counselling and screening should occur among those at-risk, with treatment strongly recommended for those who are positive (either partner). Treatment is certainly preferable to reproductive technologies such as sperm washing [15,25], but it has been demonstrated in the United States that preconception knowledge of positive status, does not necessarily lead to treatment uptake prior to pregnancy [26]. In the case of substance use, a common risk-factor for infection, counselling can incorporate the co-localization of harm-reduction approaches. Importantly, preconception planning promotes the management of risk factors and psychosocial aspects of HCV diagnosis prior to pregnancy including fear and stigma [25], while allowing clinicians and patients to collaborate on a plan of care that can include referrals to support resources and follow-up appointments. Canadian examples exist in the co-localization of trauma-informed care for women with HCV who use drugs [27], as well as the importance [28].

## 3. Prenatal Screening and Epidemiology

Current clinical practice guidelines from the Society of Obstetricians and Gynaecologists of Canada recommend universal HBsAg screening for active infection in pregnancy, but risk-based screening for HCV [18,25] (Figure 1). Of note, Saskatchewan has a long standing history of universally screening for HCV in pregnancy [29], and Alberta recently adopted this practice [30]. In comparison to Canadian guidelines, three major guideline committees now recommend universal screening every pregnancy in the United States, specifically the American Association for Study of the Liver-Infectious Disease Society of America (AASLD-IDSA) [31], the Centers for Disease Control (CDC) [32], and the American College of Obstetricians and Gynecologists (ACOG) [33]. With respect to HCV screening and diagnosis in Canada, different provinces use different algorithms. While most use reflex RNA testing after a positive antibody, Saskatchewan only reflex tests to HCV core antigen, and Ontario and Quebec do not reflex test at all [34]. HBV serology, especially in pregnancy, is more complicated to order and interpret, which may lead to a lack of familiarity among prenatal providers, especially in low endemicity countries such a Canada. Following a positive HBsAg test in pregnancy, HBeAg and HBV DNA should be completed [13,18], which is often not reflexed in Canadian provinces, and therefore may never occur.

There is a paucity of literature on the epidemiology of HBV and HCV during pregnancy among women in Canada, especially in smaller provinces. With respect to HBV, this is especially surprising considering the universal nature of HBV screening in pregnancy. The lack of epidemiology speaks to a major gap in our understanding of screening uptake, subsequent testing, and interventions to prevent transmission. In Alberta, a provincial analysis was completed reviewing all testing from 2003–2016. Over this time, tests per year increased from 45,761 to 63,475 (total samples 821,910), with a slight increase in prevalence from 0.50 to 0.58% among an estimated 95% of all pregnant persons. Trends were decreasing among younger age groups, likely owing to universal vaccination in adolescence [35]. In British Columbia, a similar uptake has been reported, with a prevalence of 0.7–1.2% [36]. We recently evaluated the prevalence in Ontario from 2012–2016. We demonstrated that while there was an overall prenatal prevalence of 0.63% among 651,745 tests performed, in regions in Northern Ontario where there has been universal birth dose vaccination for decades, prevalence is as low as 0.06–0.07%. However, in the rest of Ontario, especially where immigration is the highest in North Toronto, prevalence is anywhere from 3-fold to 25-fold higher in the absence of birth dose vaccination. Interestingly, those over the age of 45 years old had a prevalence of 2.97% [37]. We suspect this may be related to either a disproportionate number of newcomers/immigrants to Canada from HBV-endemic countries who the missed the age for universal vaccination rollout in their birth country or related to more aggressive case-finding in this age cohort due to engagement with fertility specialists.

At present there are no publicly available Canadian data on hepatitis Delta virus (HDV) co-infection among pregnant women, and HDV testing in Canada generally, may not always be completed when indicated. As such, as vertical transmission of HDV has been documented [14], HDV testing should be considered among pregnant women who are HBsAg positive, depending on the prevalence in their birth country/country of origin [38].

In the absence of universal screening in pregnancy, the true prenatal prevalence of HCV in Canada is not known. In a single study in British Columbia, among pregnant women, there was an uptake of 20.3% testing for anti-HCV among 109,983 samples, with an overall positivity rate of 2.5% [39]. Screening uptake has increased greatly and was 54.6% in 2019 [40]. More recently, we found than in the context of a universal screening pilot in 2020–2021, a prevalence of 0.4–0.55% was observed in two different urban regions in Ontario with 17,771 samples [41]. We have also looked at prevalence in Saskatchewan in the context of long-standing universal screening from 2012–2018. With approximately 15,000 births per year, there was a mean prevalence of 1.85%, ~278 births with HCV exposure [41]. Despite the difference in study findings, women of childbearing age are more likely to be at risk. Recent data from British Columbia show that rates among women born after 1974 are increasing, and this cohort is the only age group where infections among women are outpacing infections among men [42]. Importantly, from 2002 to 2014, administrative data from Ontario have demonstrated a 16-fold increase in children born to a women with opioid use disorder [43], and more recently have shown that between 2014 and 2019 one in 20 children were exposed to opioids in utero [44].

## 4. Case-Finding, Care, and Antiviral Transmission Prevention

Prenatal case-finding can maximize educational opportunities, assess liver disease, and manage care postpartum, including testing for both HBV and HCV, and post-exposure prophylaxis of the exposed infant for HBV [18,25]. Following diagnosis, linkage to further care and treatment prompts preventative care to reduce risk of transmission to others and vertical transmission in future pregnancies for HCV, and in the current pregnancy for HBV [18,25]. We will not discuss clinical interventions to avoid or use to decrease transmission (e.g., amniocentesis, c-section, reviewed in clinical practice guidelines [18,25]); rather, we will focus on virologic interventions.

Based on available data, HBV screening uptake in pregnancy likely ranges from ~80–100% depending on regional variability [35,36,37] across Canada. However, we recently showed in Ontario from 2012–2016, that following a HBsAg positive test in pregnancy, only 38% of pregnant individuals are screened for HBV DNA in the pregnancy. Based on those who were tested, we examined HBV viral load to predict the distribution of viral load among pregnant persons in Ontario, including by age. Treatment of women with an HBV DNA of >200,000 IU/mL is the standard of care in Canada [18]. We found that of 4102 HBsAg positive persons, 447 would have had a viral load high enough to warrant third trimester antivirals (Figure 1) but did not have a viral load performed in pregnancy. Although we did not link HBeAg status and HBV DNA viral load, Singh et al. have demonstrated in a Canadian context that vertical transmission is more likely to occur from HBeAg positive individuals and a high HBV viral load [45], in line with larger studies [46]; and viral load is typically stable in pregnancy [47]. Understanding why subsequent HBV testing is not completed, is an important gap in our knowledge. To date, there is no information as to whether this is because of a lack of provider knowledge in the virology, care, and management—including not being aware of the impact of third trimester antivirals; whether referrals to specialists occur, but patients are lost to follow-up for their additional specialist appointment but continue with prenatal care; or whether there are systems and social gaps leading to inadequate care such as language barriers, transportation, and access issues. Finally, it is important to emphasize that prenatal care in Canada is given by many types of providers (MDs, NPs, midwives), meaning education must occur among a very large group of providers, and further administrative data analysis of the HBV prenatal cascade of care is required.

Multiples studies have shown that use of antivirals in the third trimester of pregnancy for those with a viral load above 200,000 IU/mL can markedly reduce the risk of vaccine failure, which may be as high as 10% in children born to highly viremic mothers even if appropriately vaccinated at birth (reviewed by [48]). A landmark study from China showed that tenofovir disoproxil fumarate (TDF) reduced the risk of perinatal transmission from those with viral loads above 200,000 IU/mL from 18% to 5% (*p* = 0.007) [49]. Notably, they conservatively assumed transmission occurred among those who were lost to follow-up. However, in the group with pediatric follow-up, no transmissions occurred in the TDF group, compared to 7% in the placebo arm (*p* = 0.01). Although another trial from Vietnam showed no significant benefit of TDF over placebo, the rates of transmission were very low—0 vs. 3 of 147 births in each arm (*p* = 0.12) and importantly, this trial also documented no transmissions with the use of TDF [50]. Prior studies have evaluated lamivudine, which was shown to be effective and safe, but is not first-line therapy because of the very low barrier to resistance [50,51,52]. Indeed, in an Australian study, they found that brief exposure to lamivudine in pregnancy was associated with a high risk (19%) of developing lamivudine resistance [53]. Entecavir is not recommended in pregnancy due to possible teratogenicity in animals [54]. Most recently, the new formulation of TDF, tenofovir alafenamide (TAF) has also been studied in pregnancy. Some initial observational studies showed similar efficacy and safety to TDF, but Pan and colleagues recently reported that TAF given early in pregnancy, at 14–16 weeks—prevented transmission without the use of hepatitis B immune globulin (HBIG) at birth [55]. This study opens up the possibility of moving to earlier HBV therapy for those with high viral loads (>200,000 IU/mL) and then simplifying management in the child with vaccination, but no need for HBIG. Although less relevant in Canada, access to HBIG in many resource-constrained settings is a major limitation to HBV prevention efforts. To implement antiviral treatment in pregnancy—whether early or in the third trimester—HBV DNA testing is required, making a strong case for reflex HBV DNA in all prenatal HBsAg-positive tests.

Despite the perceived efficiency and lower cost, risk-based prenatal HCV case-finding misses opportunities for engaging women into HCV care and likely contributes to the underestimation of the prevalence of infection in pregnant individuals [40]. The main caveat of risk-based approaches being reliance on provider knowledge and patient disclosure of stigmatized risk factors necessary to classifying individuals as at-risk and requiring HCV testing. In fact, a study in Toronto clearly demonstrated that women may not be asked about, may not disclose, or may not be aware of risk factors [56]. Furthermore, pregnant women have an added disincentive to acknowledge risk activities [56], particularly injection drug use, due to stigma and fear of involvement of child protection services. We also recently determined that in a major tertiary care centre in Southwestern Ontario, that 9% of antibody positive women did not have identified risk factors [57]. Importantly, as with HBV, subsequent nucleic acid testing for HCV in the same pregnancy is poor in Ontario. Using provincial data, 56% of women received an HCV RNA one to seven years after being found antibody positive in pregnancy, of whom only 24% completed the HCV RNA within one year [41]. Follow-up HCV RNA testing rates is certainly becoming less relevant across Canada with the use of reflex RNA testing [34], however, considering the population size in Ontario and Quebec, adoption of reflex testing in these provinces will be essential to find women with active HCV infection in pregnancy.

There are currently no specific interventions during pregnancy to decrease transmission [25]. In early 2017, Canada expanded access to HCV treatments to include women planning pregnancy within the next year [58], however at present there are no recommendations regarding use of treatment during pregnancy attributed to the scarcity of human studies and trials regarding the safety and efficacy of treatment during pregnancy. In 2020, a pharmacokinetic study was published, evaluating 9 women taking sofosbuvir/ledipasvir; and found no fetal or infant growth or developmental abnormalities [59]. Similarly, an observational cohort study of 21 women taking sofosbuvir/ledipasvir during pregnancy in India, also reported that all children were born without abnormalities, and all children were negative for HCV RNA at 6 months of age [60]. The major limitation of sofosbuvir/ledipasvir is that it is only effective for HCV genotypes 1 and 4, whereas there are now pangenotypic regimens that are safe and effective for all HCV genotypes and recommended as first-line treatment in almost all settings. Pilot studies have started with sofosbuvir/velpatasvir, a pangenotypic regimen that requires 12 weeks of dosing. AASLD, suggests that even in the absence of large-scale studies, treatment in pregnancy could be considered on a case-by-case basis [31,61]. In addition to the likely maternal clinical benefit [61], women may also want to be treated in pregnancy both for their own health and the health of the baby [62]. As pregnancy and prenatal care may present a women’s only opportunity to engage in medical care, provider hesitancy towards prescribing treatment due to minimal large-scale studies of their safety and efficacy during pregnancy, needs to be weighed against the risk associated with patient loss-to-follow up, further transmission and potential development of advanced liver disease [63]. Should providers decide to treat in pregnancy (although not routinely occurring or approved in Canada), real-world data to evaluate pediatric outcomes is necessary, and as such, providers are encouraged to utilize the TiP-HepC (Treatment in Pregnancy for Hepatitis C) registry which was launched by the Centers for Disease Control and Prevention and the Coalition for Global Hepatitis Elimination in Spring 2022 [64].

## 5. Postpartum Interventions, Linkage and Pediatrics

Postpartum visits present the opportunity to engage in testing, linkage to care and treatment, and discuss safety and concerns for mothers potentially affected by HBV and HCV. Especially for those newly diagnosed in the prenatal period, close follow-up is required.

ALT flares are common post-partum in women with HBV, presumably due to the immune reconstitution that occurs after delivery. In a real-world Canadian cohort, 116 of 215 (54%) of women experienced a post-partum flare, including one that led to fulminant liver failure requiring transplantation [65,66]. Flares usually occur ~6–8 weeks post-delivery and almost always within the first 6 months [67]. A prospective study from the North American Hepatitis B Research Network, reported a lower rate (4.3%) and severity of post-partum flares. For women who start antivirals to reduce the risk of transmission, there is no clear consensus on when to stop treatment after delivery, but it is important to follow closely due to the risk of withdrawal flare, which is more common in HBeAg+ than HBeAg− women [65,68], and may occur even with extended post-partum treatment and often requires re-initiation of therapy (~20%) [68,69]. Consideration of HBV in obstetrical practices often focuses primarily on screening and then ensuring the baby receives timely birth dose vaccination and HBIG. However, it is critical to also consider the risk to the mother and thus to ensure optimal HBV care occurs both during and after delivery; and thus important for obstetrical providers to either gain experience with HBV management or refer, ideally before delivery to ensure follow-up is maintained in the post-partum period, which is both risky in terms of HBV flare and challenging due to competing priorities of a newborn.

For those diagnosed with HCV in the prenatal period, it is recommended that women follow-up postpartum for HCV care and treatment both for their own health and to prevent transmission in future pregnancies. Breastfeeding has not been linked HCV transmission and is promoted in mono-infection [25], but contraindicated in those with HIV-HCV coinfection [70]. Depending on whether a woman breastfeeds, treatment can be delayed for several months (and at times over a year) postpartum. To date there are no Canadian nor global population-level cascade of care studies for women with HCV in pregnancy, and whether they are treated postpartum (current data in Figure 2). In the Canadian context we have shown that including missed follow-up RNA testing and referrals for positive individuals, that 56% of women missed an opportunity for linkage to care during pregnancy [57]. Other groups in the United States have shown that in single-centre studies, only 9% of women are treated postpartum (10/103) [71], with larger multi-state studies modelling a follow-up postpartum rate of 6% [72].

Vertical transmission is the predominant reason for pediatric HBV and HCV, with transmission with HCV mono-infection ranging from 6–11%, compared to 11–25% in those with HIV/HCV coinfection [73,74,75]. Several factors can increase the possibility of vertical transmission including HIV coinfection, concomitant medical conditions, maternal viral load, and intravenous drug use [20,76,77]. The strongest predictor for HBV vertical transmission is HBV viral load, and this has been shown with a cohort in Alberta [66]. The chance of spontaneous clearance of acute HBV is dependent on the age of exposure. In infants and very young children, >90% of those with acute infection will progress to chronic life-long infection. However, immunocompetent adults have a clearance rate of 95% [78]. To reduce the possibility of vertical transmission, HBV viral load must be completed. Current guidelines recommend screening in each pregnancy irrespective of past immune status [13,18].

Importantly, birth dose vaccination is a major tenant of HBV elimination in Canada and globally [79]. When given within 24 h of birth, with two additional doses in infancy, HBV vaccination is 90% effective at decreasing transmission [80]. The administration of HBIG further reduces transmission. Following HBIG and birth dose vaccination, it is safe to breastfeed [18]. In endemic countries where HBV rates were high, the roll-out of birth dose vaccination has decreased prevalence from 5% to 1% in children under 5 years old, and perhaps more importantly has led to a marked decrease in the incidence of primary liver cancer [78]. Despite this, five provinces in Canada continue to vaccinate adolescents [81], and at present, the National Action Committee on Immunization recommends surveillance, with provincial changes as required [82]. However, children born in Canada are acquiring HBV prior to adolescent vaccination. Using administrative data from Ontario, we have shown that among Canadian-born children living in Ontario, there were 139 cases of HBV documented prior to adolescent vaccination. This figure is likely a significant underestimate of HBV transmission in Canadian-born children, as most children are not tested for HBV particularly given that the infection is almost always asymptomatic in early life [37]. Assuming prenatal screening was consistently high, these transmission events either came from vaccine failures, or more likely horizontal transmission events, potentially from individuals who did not know their status (another parent, grandparent, caregiver, etc.). Concerningly, in Ontario, there is no vaccination database, so we were not able to link these infections to vaccine uptake. A study from British Columbia demonstrated that among children born to HBsAg-positive mothers, 20.7% did not have a documented complete HBV vaccine series [47]. Furthermore, even when children are offered vaccination in adolescence, uptake is poor; with rates as low as 16–25% during the COVID-19 pandemic [83], and only at ~70% in Ontario pre-COVID-19 [84]. We and others have shown that HBV birth dose vaccination in Canada is cost-effective [84].

Although case-finding is likely poor as many children are never tested for infection or immunity following an exposure [47], HBV antiviral treatment is approved for use in children. However, many children in Canada who are eligible for treatment, go untreated. In a study from 2011–2018 of 325 children, 82% had an HBV DNA of >10,000 IU/mL, with ALT twice the upper limit of normal in 40% of children, with 4% with a significantly elevated ALT. Only 40 of the eligible 129 were treated [85]. Yet, liver biopsies of 134 treatment-naïve children can have significant inflammation and fibrosis [86], clear data that untreated children are at risk of liver disease.

Increased prevalence of maternal HCV has led to an increase in the proportion of infants born to HCV-infected mothers, with increases as high as 0.71% to 1.59% in Kentucky [87], with overall estimates of 0.24% in the United States [88]. The Canadian Pediatric Society currently recommends that all infants, children, and youth with one or more risk factors be screened for HCV infection [89]. Despite these guidelines, there is evidence that most children born to women with HCV are not linked to care or even tested. In Ontario, patterns of HCV screening have shown that only 29% of children born to positive mothers being tested for HCV antibody or RNA by 18 months of age [20]. Consequences of pediatric chronic HCV infection can include liver fibrosis, extrahepatic manifestations, social stigmatization, and impaired quality of life [90]. Between 20% to 30% of neonates are naturally able to clear their infection by 2 or 3 years of age [89]. Of children who do not clear their infection, infection is often mild and likely asymptomatic [91]. Importantly, there may be opportunities for low barrier testing for HCV in children, such as co-localized screening should the child be in child services, or during routine well-baby/immunization visits. In the past, there may not have been the impetus to screening in children in the absence of treatment. However, treatment is now available in Canada as young as age 3 [92], strengthening the rationale to ensure children are screened, diagnosed, and linked to care as early as possible.

## 6. Opportunities for Elimination: How to Better Serve Pregnant Persons and Children with HBV/HCV

As Canada strives towards the WHO goals for the elimination of viral hepatitis as a public health threat by 2030, significant progress can be made by intensifying efforts to expand testing and care to pregnant individuals and children. We have summarized considerations and recommendations for Canada to achieve this goal in this population (Figure 3). For both HBV and HCV, the preconception period provides the opportunity to engage individuals and their partners in testing, treatment considerations, and counselling to ensure the infection is managed and risks of complications during pregnancy and postpartum are reduced. Universal routine prenatal screening for HBV is already the standard of care in Canada, but subsequent HBeAg and viral load testing is inadequate, which could be improved with the introduction of HBeAg and/or HBV DNA reflex testing. Only then would all eligible individuals be offered antivirals in pregnancy which would greatly reduce transmission in the context of a high viral load. Universal HCV screening in pregnancy could maximize opportunities for case-finding and subsequent linkage to care for both mothers and their newborns, but reflex HCV RNA testing would also need to be in place to maximize efforts. For HCV, further consideration regarding the use of antiviral treatments during pregnancy and patient preference should be weighed as an option against the risk of loss to follow-up. All possible interventions should occur in the postpartum period to treat pregnant women with HCV to prevent transmission in future pregnancies, and improved screening linkage to care and treatment for children with HCV is essential. HBV vertical transmission is entirely preventable, and yet, Canadian-born children are acquiring HBV, which can lead to serious health consequences. Universal birth dose vaccination and better follow-up of children will be the only way to achieve the elimination of new infections among children in Canada. Finally, a better understanding of the cascades of care for both HBV and HCV including administrative data evaluation and linkage will be essential to track progress towards micro-elimination in this population.

## Figures and Tables

**Figure 1 viruses-15-00091-f001:**
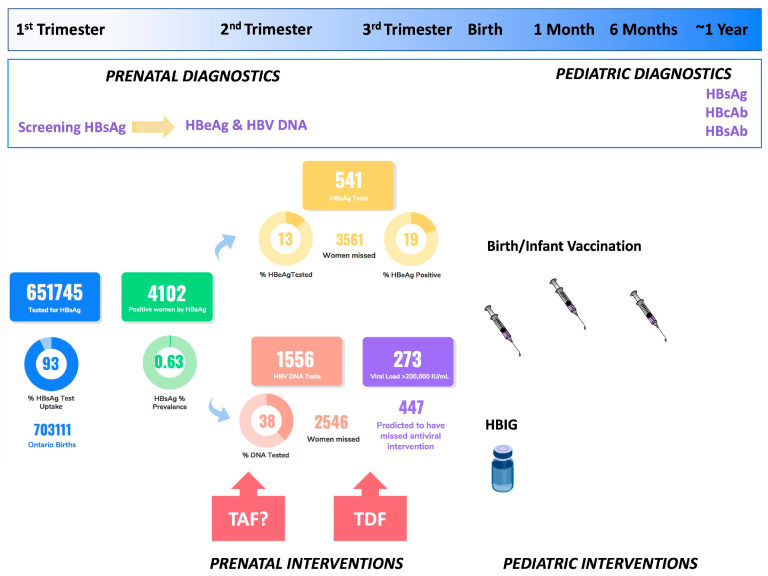
HBV prenatal screening uptake, prevalence, and missed opportunities for interventions from 2012–2016 (reproduced from [37] with modifications), and interventions to prevent transmission.

**Figure 2 viruses-15-00091-f002:**
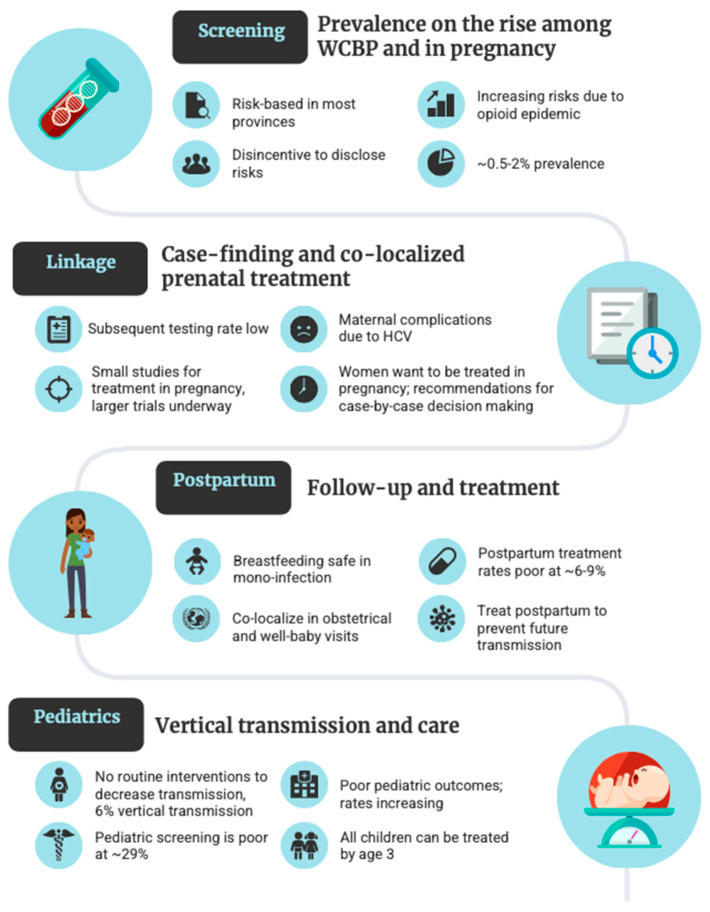
Canadian and global data on the steps in the HCV cascade of care.

**Figure 3 viruses-15-00091-f003:**
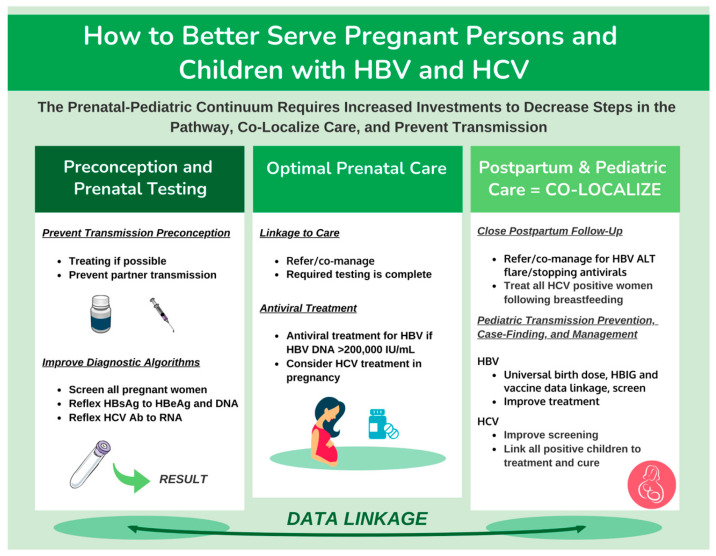
Opportunities towards the elimination of HBV and HCV in pregnant persons and children.

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
