# Peer review of "Hepatitis B and C in Pregnancy and Children: A Canadian Perspective"

_viruses, 2022, doi:10.3390/v15010091_

Round 1

Reviewer 1 Report

Well written and commented, comprehensive review article on pre-post conception and delivery management of viral hepatitis B & C,that looking at the local epidemiology of VH and related PH recommendations, has a strong translational commitment.

Only minor comments:

Owing to the burden of migrants and ethnics in Canada, I think it would be worth empowering the section of HBV management  with the recommendation of HDV screening among pregnant persons.

I am not sure that references 2,12,30,29,36,40,41,57 and 84 match the editorial requirements.

Author Response

Dear Reviewer 1, 

We thank you for this feedback. 

1. Owing to the burden of migrants and ethnics in Canada, I think it would be worth empowering the section of HBV management  with the recommendation of HDV screening among pregnant persons. Response: We have added the following statement "At present there is no publicly available Canadian data on hepatitis Delta virus (HDV) co-infection among pregnant women, and HDV testing in Canada generally, may not always be completed when indicated. As such, as vertical transmission of HDV has occurred14, HDV testing should be considered among pregnant women who are HBsAg positive, depending on the prevalence in their birth country/country of origin38".

2. I am not sure that references 2,12,30,29,36,40,41,57 and 84 match the editorial requirements. Response: We have modified to use a different output style in the revised version. 

Reviewer 2 Report

Mendlowitz et al have submitted an excellent review of the current prevalence of HBV/HCV in those of child bearing age in Canada and have offered insightful solutions to improving management and decreasing childhood hepatitis. 

1. You state that there are no recommendations in a pre-conception population.  Is it the same for individuals seeking assistance with pregnancy (like IVF) as it would be for a couple not seeking reproductive assistance?

2.  Prenatal screening:  might expand on why the prevalence of HBsAg in those over 45 is interesting.

3.  Might you speculate on why HBsAg testing is so well adopted but HBV DNA testing is not?  does it require a referral to a subspecialist or is OB/GYN comfortable with this algorithm?

4. Do you really want to reflex HBsAg to both eAg and DNA?  how is eAg status going to change your management?  

5.  If you are advocating for HCV therapy during pregnancy would also recommend they add to the data to the Global Task force database (or some database) 

Author Response

Dear Reviewer 2, 

We thank you for this feedback.

1. You state that there are no recommendations in a pre-conception population.  Is it the same for individuals seeking assistance with pregnancy (like IVF) as it would be for a couple not seeking reproductive assistance? Response: We have removed the sweeping statement that there are not recommendations beyond clinical care, we were speaking of Canadian-specific, but can appreciate that that globally that statement is not correct, as all interventions such as partner vaccinations are "recommendations". As per ACOG, if the sperm donor is positive, sperm washing + IUI or IVF does not reduce transmission. We have modified to "Once the partner is immune, no reproductive assistance processes are required, such as sperm washing14, as pursuing intrauterine insemination (IUI) or in vitro fertilization (IVF) has not been shown to decrease transmission15. We also re-organized the structure so the section on preconception treatment follows, which is another potential intervention pre-conception.  

2. Prenatal screening:  might expand on why the prevalence of HBsAg in those over 45 is interesting. Response: We have added the following "We suspect this may be related to either a disproportionate number of newcomers/immigrants to Canada from HBV-endemic countries who the missed the age for universal vaccination rollout in their birth country, or related to more aggressive case-finding in this age cohort due to engagement with fertility specialists"

3. Might you speculate on why HBsAg testing is so well adopted but HBV DNA testing is not?  does it require a referral to a subspecialist or is OB/GYN comfortable with this algorithm? Response: Excellent point. We have added the following "Understanding why subsequent HBV testing is not completed, is an important gap in our knowledge. To date, there is no information as to whether this is because of a lack of provider knowledge in the virology, care, and management including not being aware of the impact of third trimester antivirals; whether referrals to specialists occur, but patients are lost to follow-up for their additional specialist appointment but continue with prenatal care; or whether there are systems and social gaps in our ability to care for these patients leading to inadequate care such as language barriers, transportation, and access issues. Finally, it is important to emphasize that prenatal care in Canada is provided by many types of providers (MDs, NPs, midwives), meaning education must occur among a very large group of providers, and further administrative data analysis of the HBV prenatal cascade of care is required."

4. Do you really want to reflex HBsAg to both eAg and DNA?  how is eAg status going to change your management? Response: Although it may not change management to know both, what we have learned across provinces for the introduction of HCV RNA reflex testing (nucleic acid based) from HCV antibody testing (serology based) is either due to workflow or lack of understanding of nucleic acid stability, laboratories have delayed the implementation of serology to nucleic acid reflex testing. Thus, we would like to keep this statement open in the context of HBV to reflect the fact that having one would be superior to having neither. We have changed this sentence in the discussion to: "Universal routine prenatal screening for HBV is already the standard of care in Canada, but subsequent HBeAg and viral load testing is inadequate, which could be improved with the introduction of HBeAg and/or HBV DNA reflex testing."

5. If you are advocating for HCV therapy during pregnancy would also recommend they add to the data to the Global Task force database (or some database). Response: We are not necessarily advocating to adopt a similar approach to the US, as there are many significant differences in our healthcare systems. Of particular relevance is the fact that women in the US loose their medication coverage 6 weeks postpartum, while in Canada that is not the case, i.e. "For HCV, further consideration regarding the use of antiviral treatments during pregnancy and patient preference should be weighed as an option against the risk of loss to follow-up." Nonetheless, the database mentioned is important to collect global data, which could affect Canadian decision-making in the future. As such, we have added the following sentence "Should providers decide to treat in pregnancy (although not routinely occurring or approved in Canada), real-world data to evaluate pediatric outcomes is necessary, and as such, providers are encouraged to utilize the TiP-HepC (Treatment in Pregnancy for Hepatitis C) registry which was launched by the Centers for Disease Control and Prevention and the Coalition for Global Hepatitis Elimination in Spring 202265."